# Lung Ultrasound Patterns and Clinical-Laboratory Correlates during COVID-19 Pneumonia: A Retrospective Study from North East Italy

**DOI:** 10.3390/jcm10061288

**Published:** 2021-03-20

**Authors:** Riccardo Senter, Federico Capone, Stefano Pasqualin, Lorenzo Cerruti, Leonardo Molinari, Elisa Fonte Basso, Nicolò Zanforlin, Lorenzo Previato, Alessandro Toffolon, Caterina Sensi, Gaetano Arcidiacono, Davide Gorgi, Renato Ippolito, Enrico Nessi, Pietro Pettenella, Andrea Cellini, Claudio Fossa, Eleonora Vania, Samuele Gardin, Andi Sukthi, Dora Luise, Maria Teresa Giordani, Mirko Zanatta, Sandro Savino, Vito Cianci, Andrea Sattin, Anna Maria, Andrea Vianello, Raffaele Pesavento, Sandro Giannini, Angelo Avogaro, Roberto Vettor, Gian Paolo Fadini, Alois Saller

**Affiliations:** 1Department of Medicine, University of Padova, Via VIII Febbraio, 2, 35122 Padova, Italy; caponefederico@hotmail.com (F.C.); lorenzo.cerruti1991@gmail.com (L.C.); leonardo.molinari@tiscali.it (L.M.); lorenzo.previato@unipd.it (L.P.); Ale.toffolon@gmail.com (A.T.); caterushi@hotmail.it (C.S.); gparcidiacono91@gmail.com (G.A.); davide.gorgi92@gmail.com (D.G.); sandro.savino@unipd.it (S.S.); sandro.giannini@unipd.it (S.G.); angelo.avogaro@unipd.it (A.A.); roberto.vettor@unipd.it (R.V.); gianpaolofadini@hotmail.com (G.P.F.); alois.saller@unipd.it (A.S.); 2Department of Emergency, University Hospital of Padova, Via Giustiniani, 2, 35128 Padova, Italy; Stefano_pasqualin@libero.it (S.P.); Elisafb90@gmail.com (E.F.B.); nicolozanforlin@gmail.com (N.Z.); renato.ippolito@aopd.veneto.it (R.I.); enrico.nessi89@gmail.com (E.N.); pietro.pettenella@gmail.com (P.P.); andrea.cellini@aopd.veneto.it (A.C.); claudio.fossa@aopd.veneto.it (C.F.); eleovn9@gmail.com (E.V.); vito.cianci@aopd.veneto.it (V.C.); 3Department of Infectious Diseases, University Hospital of Padova, Via Giustiniani, 2, 35128 Padova, Italy; samuele.gardin@studenti.unipd.it (S.G.); andrea.sattin@aopd.veneto.it (A.S.); annamaria.cattelan@aopd.veneto.it (A.M.); 4Department of Cardiac, Thoracic, Vascular Sciences and Public Health, University of Padova, Via Giustiniani, 2, 35128 Padova, Italy; andi.sukthi@aopd.veneto.it (A.S.); andrea.vianello@aopd.veneto.it (A.V.); 5Infectious and Tropical Diseases, San Bortolo Hospital, Viale Rodolfi, 37, 36100 Vicenza, Italy; luise.dora@gmail.com (D.L.); mt.giordani@aulss8.veneto.it (M.T.G.); 6Department of Emergency, Arzignano Hospital, Via del Parco, 1, 36071 Arzignano, Italy; mirko.zanatta@aulss8.veneto.it; 7Internal Medicine, Montebelluna Hospital, Via Palmiro Togliatti, 1, 31044 Montebelluna, Italy; raffaele@pesavento.eu

**Keywords:** COVID-19, lung ultrasound, B-lines, consolidations

## Abstract

Background and Aim. Lung ultrasound (LUS) is a convenient imaging modality in the setting of coronavirus disease-19 (COVID-19) because it is easily available, can be performed bedside and repeated over time. We herein examined LUS patterns in relation to disease severity and disease stage among patients with COVID-19 pneumonia. Methods. We performed a retrospective case series analysis of patients with confirmed SARS-CoV-2 infection who were admitted to the hospital because of pneumonia. We recorded history, clinical parameters and medications. LUS was performed and scored in a standardized fashion by experienced operators, with evaluation of up to 12 lung fields, reporting especially on B-lines and consolidations. Results. We included 96 patients, 58.3% men, with a mean age of 65.9 years. Patients with a high-risk quick COVID-19 severity index (qCSI) were older and had worse outcomes, especially for the need for high-flow oxygen. B-lines and consolidations were located mainly in the lower posterior lung fields. LUS patterns for B-lines and consolidations were significantly worse in all lung fields among patients with high versus low qCSI. B-lines and consolidations were worse in the intermediate disease stage, from day 7 to 13 after onset of symptoms. While consolidations correlated more with inflammatory biomarkers, B-lines correlated more with end-organ damage, including extrapulmonary involvement. Conclusions. LUS patterns provide a comprehensive evaluation of patients with COVID-19 pneumonia that correlated with severity and dynamically reflect disease stage. LUS patterns may reflect different pathophysiological processes related to inflammation or tissue damage; consolidations may represent a more specific sign of localized disease, whereas B-lines seem to be also dependent upon generalized illness due to SARS-CoV-2 infection.

## 1. Introduction

Coronavirus disease 2019 (COVID-19) primarily hits the respiratory system and can secondarily involve extrapulmonary organs, such as the heart, liver, kidneys, pancreas, and brain [1]. SARS-CoV-2 (severe acute respiratory syndrome–coronavirus 2) enters the respiratory tract mainly via Angiotensin Converting Enzyme 2 (ACE2) and replicates in airway epithelial cells [2]. The spectrum of respiratory manifestations of COVID-19 ranges from asymptomatic to mild upper airway cold-like symptoms, to lower airway symptoms, such as cough and dyspnea. The most severe manifestation is interstitial pneumonia, typically bilateral, which leads to respiratory failure needing invasive ventilation [3].

Chest X-ray has limited sensitivity for the diagnosis of COVID-19 pneumonia, and no specific signs can be detected. Computed tomography (CT) is considered the gold standard for the diagnosis of interstitial pneumonia but may not be universally available and cannot be performed bedside, thereby posing problems for the repeated examination of patients isolated in COVID-19 wards [4].

Lung ultrasound (LUS) represents a suitable imaging alternative to both chest X-ray and CT scan for COVID-19 diagnosis and monitoring because it can be performed bedside (even at home) by an internist, can be easily repeated over time, and does not require sophisticated instruments or radiation [5].

Several studies have examined the potential use of LUS in COVID-19 patients. So far, for the diagnosis of COVID-19, CT scan outperforms LUS, which is sensitive and correlates with CT findings [6,7], but has limited specificity [8]. No study so far has described in detail the pattern of LUS findings in patients with COVID-19 interstitial pneumonia in relation to disease severity, stage, and other clinical and laboratory features. Prior studies were small (*n* = 10–20 patients) and provided no or limited data on clinical-laboratory correlates [9,10,11]. Moreover, the different theoretical meaning of the two most common ultrasound patterns, B-lines and consolidations, has not been thoroughly addressed. Most LUS scores developed so far do not consider B-lines and consolidations independently, but as part of the same pathological process [12,13]. However, B-lines are known to be associated not only with inflammation, but also with pulmonary edema, while consolidations are not. Therefore, examining the clinical and pathophysiological correlates of different LUS patterns is of great interest.

Recently, a simple severity index has been developed and validated to identify patients who will progress to respiratory failure within 24 h, and it is based on a few routinely available measures [14]. This quick COVID-19 severity index (qCSI) outperforms the Elixhauser mortality index, CURB-65, and quick SOFA (Sepsis-related Organ Failure Assessment), therefore, it represents the best prognostic index so far available.

We herein retrospectively examined patterns of LUS in patients with pneumonia and confirmed SARS-CoV-2 infection in relation to severity as defined by the qCSI, disease stage as defined by days elapsed since onset of symptoms, and in relation to several clinical-laboratory features.

## 2. Materials and Methods

### 2.1. Patient Population and Data Collection

We conducted a multicenter retrospective cohort study, enrolling patients from three hospitals in the Veneto region, North East Italy. Patients were recruited at two Emergency Departments (University Hospital of Padua and Arzignano Hospital), three non-intensive COVID-19 wards (the infectious disease ward and COVID-19 internal medicine ward of the University Hospital of Padua, and the infectious disease ward of Vicenza Hospital) and the pneumological sub-intensive care unit of the University Hospital of Padua. All consecutive patients with laboratory-confirmed SARS-CoV-2 infection as defined by WHO (positive results of real-time reverse transcription polymerase chain reaction assay of nasal and pharyngeal swabs) [15] and clinical/radiological signs of acute COVID-19 pneumonia admitted from 26 February to 6 April 2020 were considered for enrollment. Exclusion criteria were inability to collaborate in the execution of LUS and lack of microbiologic confirmation of SARS-CoV-2 infection during hospitalization.

We collected demographic data, comorbid conditions, medications, physical examination, and laboratory findings on admission and at the time of LUS execution. Patients were risk-stratified according to qCSI [14] and time from the onset of symptoms. Stage 1 (early disease) was defined from onset of symptoms to day 6; stage 2 (acute phase) was defined from day 7 to 13; stage 3 (resolving phase) was defined from day 14 on. Clinical follow-up was obtained by daily review of all medical records. Outcome analysis started at time of baseline LUS exam.

Study data were collected anonymously and managed by the medical staff using REDCap electronic data capture tools hosted at the University of Padova [16]. All clinical investigations were conducted in compliance with the Declaration of Helsinki (2001). In agreement with local and national regulations on retrospective studies using anonymized data, the need for informed consent was waived and the study protocol was notified by the local Ethics committee of the participating centers.

### 2.2. Ultrasound Examination and Scoring

The LUS team was composed of operators with long-standing experience in LUS, who had been involved in the clinical management of COVID-19 since the beginning of the pandemic. The team standardized the equipment and acquisition protocol. Operators were dressed in protective clothing, gloves and goggles to enter COVID-19 areas. All precautions for respiratory, droplet and contact isolation were provided. All studies were performed bedside in the designated COVID-19 areas using dedicated scanners. For acquiring data, the patient’s chest was divided and scanned in twelve different areas (2 anterior, 2 lateral, and 2 posterior for each lung) on the basis of anatomic landmarks (Appendix A and Figure 1). Every area was imaged for 10 s with a single intercostal scan. We used convex or linear transducers, according to the patient’s body size. We kept the mechanical index as low as possible (starting from 0.7 and reducing it further if allowed by the visual findings). To avoid saturation phenomena as much as possible, we controlled gain and diminished the mechanical index as needed. When LUS was performed on patients unable to maintain the sitting position, the operator obtained partial views. If the quality of the acquisition was insufficient, the area was excluded from the analysis. A numeric score ranging from 0 to 3 was assigned to each area depending on severity of the findings for each of the following items: B-lines, lung consolidation, pleural line integrity, pleural effusion (Appendix A). The scores were meant as discrete steps of a continuous severity scale. For a simplified representation of LUS patterns, we averaged the cumulative scores of the right and left lung within defined patient subgroups. For each area, we also recorded the possible presence of pneumothorax (absence of pleural sliding).

To minimize inter-observer variability in scoring assignment, before starting data collection, operators from each center shared a total of 10 cases of confirmed COVID-19 in an anonymized virtual database. All frames were acquired by video clips of 5 min duration, where the lung surface below the landmark could be monitored and analyzed for a few seconds during breathing. Along with the video clip, the operator provided the score for each item of every area. Every operator discussed his clinical cases with the LUS COVID team through online meetings. Video clips were discussed by all team members, who were blind to the clinical background, and the assigned score were reviewed. Data collection began when the scoring assignment was considered homogeneous for each operator.

### 2.3. Outcomes

The aim of this study was to investigate the usefulness of LUS in assessing COVID-19 severity and to identify potential correlates between LUS patterns and stages of the disease as well as other clinical-biochemical features. The primary outcome was the association between LUS patterns and COVID-19 severity index according to qCSI [14]. Secondary outcomes were to evaluate the association between LUS patterns and disease stage according to time from onset of symptoms, as well as the association of total LUS score with other clinical-laboratory features of COVID-19.

### 2.4. Statistical Analysis

Data are presented as the mean and standard deviation for continuous variables or as a percentage for categorical variables. Comparison of continuous variables between two or more groups were performed using the unpaired two-sided Student’s *t* test and ANOVA, respectively. Comparison of categorical variables was performed using the chi square test. Linear correlations were performed using the Pearson’s r coefficient. In case of skewed variables in the Shapiro–Wilk test, data were log-transformed before analysis with parametric tests. SPSS software version 26.0 (IBM, Armonk, NY, United States) was used for statistical analysis. Statistical significance was accepted at *p* < 0.05.

## 3. Results

### 3.1. Patient Characteristics

The initial population included 307 patients examined at participating centers. After excluding patients for whom LUS was not performed according to the standardized protocol (*n* = 149) or the score could not be computed due to incompleteness of the analysis (*n* = 62), 96 patients were included in the study. They were on average 65.9-year-old, 58.3% were males and had high prevalence of cardiovascular risk factors, such as diabetes (24%), hypertension (42.7%), obesity (27.1%) and dyslipidemia (19.8%). Prevalence of active smoking was 6.3% and only 10.4% had a prior diagnosis of chronic obstructive pulmonary disease (COPD). Patients received all medications that were, at each given time, supposed to provide benefits, including hydroxychloroquine, tocilizumab, anti-virals, antibiotics, glucocorticoids, and heparin.

Patients with a high qCSI (>3), as compared to those with a low qCSI, were significantly older, had worse respiratory parameters and inflammatory biomarkers, but showed no other significant difference in terms of history and therapies. The outcome of patients with high qCSI was worse, with significantly more frequent need for high-flow oxygen and invasive or non-invasive ventilation (Table 1).

### 3.2. LUS Patterns and COVID-19 Severity

Herein, we focused on the analysis of LUS patterns related to the two most typical features of COVID-19 pneumonia, namely, B-lines and consolidations. For each of the six lung fields examined with LUS, we averaged the sum of scores obtained in the right and left lungs and represented them graphically, as depicted in Figure 1A, using the color code to show the severity of B-lines and consolidations. Geographical LUS patterns were clearly different in the two groups. As expected, B-lines were found mainly in the basal posterior fields and were much less common in the anterior fields, especially in patients with low qCSI. In patients with high qCSI, the score for B-lines was significantly higher than in patients with low severity index for all fields except the upper posterior, while the greatest difference was observed for the anterior fields. Similarly, consolidations were most commonly observed in the basal posterior fields irrespective of qCSI. In patients with high versus low qCSI, the consolidation score was significantly 2 to 4 times higher for all lung fields (Figure 1B).

Both the B-line and consolidation scores were able to discriminate patients with severe COVID-19 and their performance, based on the area under receiver operating characteristics curve, was similar (0.71; 95% C.I. 0.62–0.81 and 0.69; 95% C.I. 0.58–0.79, respectively; Figure 1C).

### 3.3. LUS Patterns According to Disease Stage

We then analyzed LUS patterns in patients divided according to disease stage, which was defined by the time elapsed since onset of symptoms. Geographical distribution of the B-line score was significantly different across the three disease stages. Particularly, B-line scores increased significantly in the anterior fields and upper lateral field in the intermediate disease stage (from day 7 to 13 after onset of symptoms) compared to stage 1 (early after onset of symptoms), while it was not significantly different in the other fields over time. Interestingly, B-line scores were quite similar in early (stage 1) and late (stage 3) COVID-19. The geographic pattern of consolidations did not change significantly across the three disease stages (Figure 2A).

To evaluate how LUS score changed over time in the two groups defined by qCSI, we computed the average total LUS score for B-lines and consolidations by summing scores of all fields in both lungs. Total B-line score was >2-times higher in stage 1 among patients with high qCSI compared to those with low qCSI. The total B-line score worsened from stage 1 to stage 2 among patients with low qCSI and became no different from that observed among patients with high qCSI. The total score for consolidations remained higher among patients with high qCSI compared to those with low qCSI, a difference that was significant at stage 1 and 2 (Figure 2B).

### 3.4. Clinical Correlates of LUS Score

Finally, we used the total LUS score computed as described above to evaluate the clinical-laboratory correlates of B-lines and consolidations. We retained only variables that showed at least one statistically significant correlation with B-line or consolidation scores, according to the conventional 5% type 1 error (Figure 3). There were several significant positive or negative correlations of LUS scores for B-lines and consolidations with markers of respiratory function, hemato-inflammatory activation, and organ damage. Among markers of respiratory function, respiratory rate was that most associated with LUS findings, showing a direct, significant correlation with both B-lines and consolidation scores. Among hemato-inflammatory markers, there were different patterns of correlations, with C-reactive protein and platelet count being directly correlated with consolidation scores on the one side, and lymphocyte count being inversely correlated with B-line scores on the other side. Among markers of organ damage, LDH, AST and troponin I correlated directly with the B-line score, whereas troponin I and creatinine correlated with the consolidation score. Except for respiratory rate, only concentrations of D-dimer were significantly and consistently correlated with both B-line and consolidation scores.

## 4. Discussion

Using a uniform protocol for performing and reporting LUS, we were able to describe geographic and severity patterns in patients admitted for COVID-19 pneumonia. B-lines and consolidations were most common in the posterior inferior fields over the entire course of disease. By combining data from different patients, we could describe the changes in LUS patterns at three disease stages. Interestingly, the worst LUS patterns, for both B-lines and consolidations, were clearly observed in stage 2, i.e., in the second week after onset of symptoms, when disease manifestations are supposed to be sustained by a forward feeding inflammatory state and “cytokine storm” [17]. In the early disease stage (first week of symptoms) and in the later phase, when resolution starts, LUS patterns for B-lines and consolidations were similar. Thus, this finding clearly supports the use of LUS for monitoring the evolution of COVID-19 pneumonia, possibly by repeating the exam over time. For example, a prior small case series (*n* = 10) reported feasibility of repeating LUS to monitor COVID-19 pneumonia progression in patients with respiratory failure [18]. Interestingly, LUS findings can also entail prognostic implications and guide clinical management [19].

In addition, we were able to identify LUS patterns associated with severity as defined by the qCSI. qCSI is a simple severity score calculated at admission from readily available clinical parameters and is so far considered the best indicator of COVID-19 outcome, predicting rapid deterioration and need of intensive care [14]. We found dramatically worse LUS scores for B-lines and consolidations in the vast majority of lung fields of patients with high versus those with low qCSI. B-line and consolidation scores were equally able to discriminate patients with high qCSI, who are expected to progress to severe COVID-19. This finding supports the use of bedside LUS for complementing the clinical evaluation of COVID-19 severity [20]. We also identified the differential associations between LUS findings and clinical-laboratory features of the entire spectrum of COVID-19 manifestations, such as respiratory dysfunction, hemato-inflammatory activation, and extrapulmonary organ damage. Interestingly, while consolidations were correlated to inflammatory biomarkers, B-lines were more strongly correlated with end-organ damage. This finding gives a pathophysiological basis for LUS patterns because consolidations may represent a more specific sign of localized disease, possibly indicating bacterial superinfection unrelated to SARS-CoV-2, whereas the worsening B-line score may be more directly related to systemic spreading of the viral infection and organ damage due to COVID-19. The observation that B-lines are a less specific sign of pneumonia than consolidations is consistent with the clinical practice of bedside lung ultrasound assessment. In fact, except for distribution, B-lines due to viral inflammation cannot be easily distinguished from B-lines due to volume overload and pulmonary edema, which are commonly observed in critically ill patients with multiorgan failure. Therefore, severely ill COVID-19 patients may present with a high number of B-lines partly due to cardiac involvement, resulting in fluid retention in the lungs. Besides, creatinine concentrations were correlated with consolidations but not with B-lines, suggesting that hypovolemic status was poorly related to B-lines in these patients. This observation suggests that lung ultrasound in these patients does not just represent pulmonary damage, but likely reveals systemic events occurring during SARS-CoV-2 infection.

Pulmonary thromboembolism is a common complication of COVID-19 patients. However, pulmonary embolism is not usually associated with pathological lung ultrasound findings. So, we believe that lung ultrasound is of limited value for diagnosing this COVID-19 complication in our patients, all of whom had pneumonia with some degree of lung ultrasound pathological findings. Yet, based on these considerations, we suggest that pulmonary embolism should be suspected in COVID-19 patients with severe respiratory failure and only mild lung ultrasound alterations. In this setting, bedside ultrasonography may be especially helpful to rule out lower limb venous thrombosis with whole-leg compression ultrasound. Ferritin concentrations, a well-known inflammatory marker repeatedly shown to be elevated in severe COVID-19 [21,22], showed a paradoxical inverse correlation with lung consolidations. We speculate that ferritin elevation may mainly reflect systemic inflammation rather than a local pulmonary process.

Although we recognize that the literature on COVID-19 is expanding at an unprecedented fast pace, to the best of our knowledge, there is a critical lack of information on quantitative and qualitative LUS patterns in sufficiently representative cohorts of patients with COVID-19 pneumonia. Thus, our study helps to fill such a knowledge gap by providing new important information on the LUS findings that can be expected in patients with COVID-19 with various severity and at different stages of the disease. Other strengths of the present study include the standardization of the LUS protocol, with a uniform data collection form used by all trained operators, which allows pooling and comparing the data. This point is greatly relevant, as results of ultrasound examinations are often considered to be strongly operator-dependent especially in a challenging scenario, such as that of isolated COVID-19 wards. In addition, we performed an extensive clinical and laboratory characterization of patients included in the analysis, which allowed us to detect specific features related to LUS patterns.

The study has, however, some limitations. The routine care setting wherein it was performed necessarily introduces some variability in patients’ care. In addition, we reconstructed COVID-19 progression through stages using data from different patients while this should be ideally done by serial imaging of the same patients. Therapies administered during the hospitalization were highly variable, also in relation to the evolving evidence during the pandemic [23]. Therefore, it is possible that part of the LUS changes observed according to disease stage and severity were due to in-hospital therapies, even though most have so far proven ineffective (except for glucocorticoids) [24,25,26,27].

LUS has several advantages over chest X-ray and CT scan, due to its wide availability and usability at the patient’s bedside [4]. Description of the specific LUS patterns according to disease stage and severity is important for continuing exploration of this imaging modality in COVID-19 patients.

## 5. Conclusions

Our data show how LUS can provide a comprehensive picture of the evaluation of COVID-19 pneumonia and how it correlates with severity and reflects disease stage. Moreover, LUS patterns may reflect different pathophysiological processes related to inflammation or tissue damage. Specifically, consolidations may represent a more specific sign of localized disease, whereas B-lines seem to be dependent upon generalized illness due to SARS-CoV-2 infection. Thus, LUS is a promising tool, not only for evaluating the lungs, but also for a comprehensive assessment of the SARS-CoV-2 infection.

Further studies are needed to confirm its clinical usefulness in patient management, risk stratification and therapeutic monitoring.

## Figures and Tables

**Figure 1 jcm-10-01288-f001:**
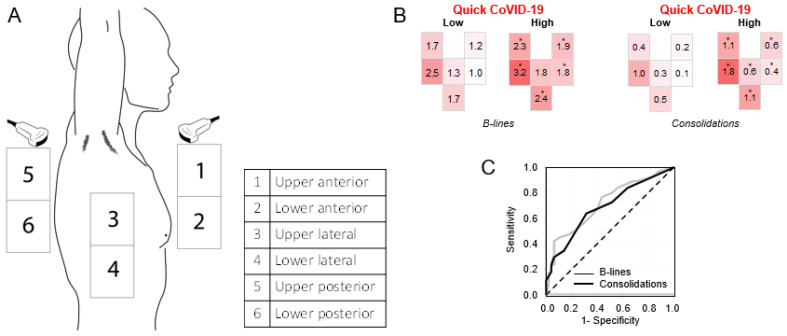
Graphical representation of lung ultrasound (LUS) patterns. (**A**) Geographical location of the six lung fields. (**B**) Average score for B-lines and consolidation in each of the lung field in patients divided according to high or low quick coronavirus disease 2019 (COVID-19) severity index (* *p* < 0.05 high versus low). (**C**) Receiver Operating Characteristics (ROC) curve for the discrimination of high quick COVID-19 severity index (qCSI) by B-line and consolidation scores.

**Figure 2 jcm-10-01288-f002:**
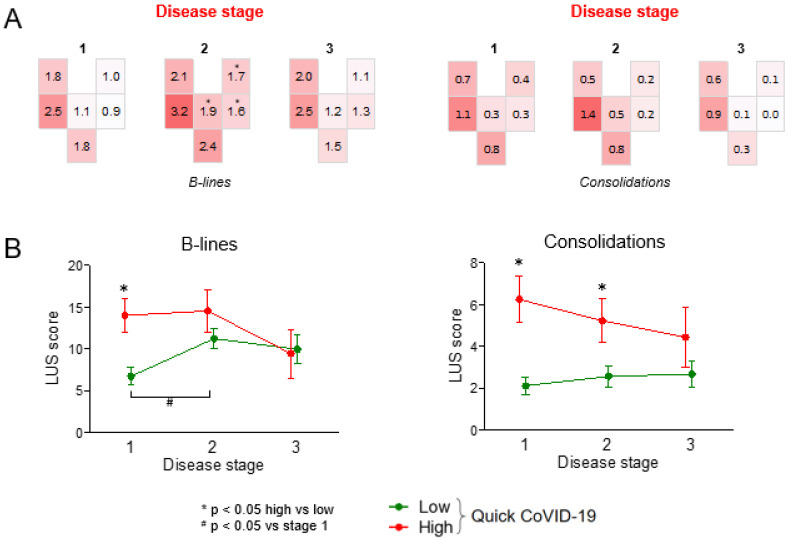
LUS patterns according to disease stage. (**A**) Geographical LUS patterns according to disease stage (1: day 0–6; 2: day 7–13; 3: day 14+). * *p* < 0.05 versus stage 1. (**B**) Time trend of the total B-line and consolidation score in the two groups of patients divided according to the quick COVID-19 severity index (* *p* < 0.05 high vs. low; # *p* < 0.05 vs. stage 1).

**Figure 3 jcm-10-01288-f003:**
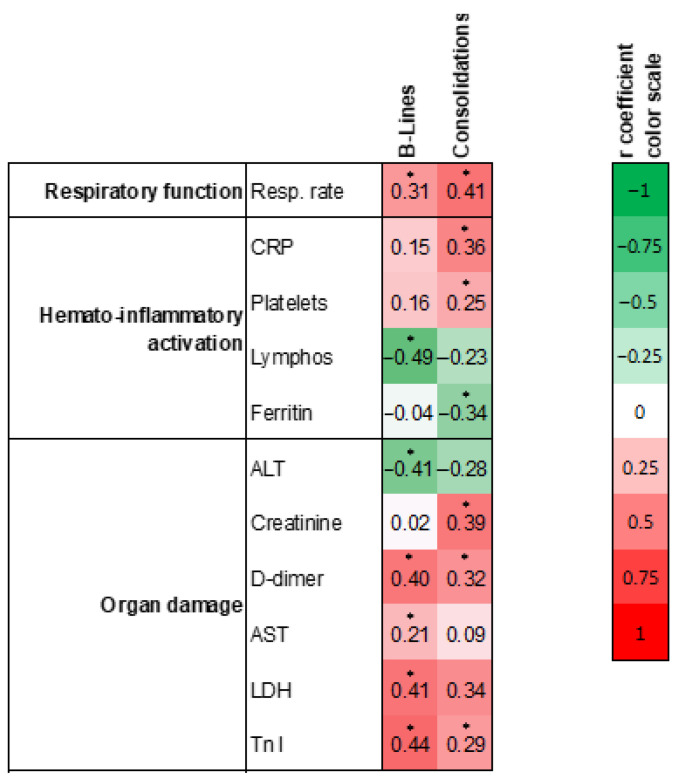
LUS scores and clinical-laboratory parameters. Clinical and laboratory parameters were divided into those related to respiratory function, hemato-inflammatory activation, and end-organ damage. Numbers indicate regression coefficients and * indicate statistical significance (*p* < 0.05). The color code shown on the right reflects sign and degree of the correlation. CRP; C-reactive protein; ALT: alanine aminotransferase; AST: aspartate aminotransferase; LDH: lactate dehydrogenase; Tnl: Troponin I.

**Table 1 jcm-10-01288-t001:** Characteristics of the patients distinguished by qCSI.

	All (*n* = 96)	qCSI ≤ 3 (*n* = 69)	qCSI > 3 (*n* = 27)	*p* Value
Demographics				
Age, mean (SD), years	65.87 (15.28)	63.56 (15.86)	71.77 (12.04)	0.008
Male, *n* (%)	56 (58.33)	38 (55.07)	18 (66.67)	0.300
Female, *n* (%)	40 (41.67)	31 (44.93)	9 (33.33)	0.300
Ethnicity, % Caucasian	89 (93.68)	64 (94.12)	25 (92.59)	0.782
Weight, mean (SD), kg	78.37 (14.71)	80.05 (15.24)	73.25 (11.94)	0.074
Height, mean (SD), m	1.68 (0.22)	1.67 (0.26)	1.69 (0.07)	0.661
Comorbidities				
Hypertension, *n* (%)	41 (42.71)	31 (44.93)	10 (37.04)	0.482
Diabetes, *n* (%)	23 (23.96)	19 (27.54)	4 (14.81)	0.189
Smoking, *n* (%)	6 (6.25)	4 (5.8)	2 (7.41)	0.769
BMI > 30, *n* (%)	26 (27.08)	22 (31.88)	4 (14.81)	0.090
Hyperlipidemia, *n* (%)	19 (19.79)	13 (18.84)	6 (22.22)	0.708
Coronary disease, *n* (%)	5 (5.21)	2 (2.9)	3 (11.11)	0.103
TIA/Stroke, *n* (%)	6 (6.25)	4 (5.8)	2 (7.41)	0.769
Heart failure, *n* (%)	4 (4.17)	3 (4.35)	1 (3.7)	0.887
Atrial Fibrillation, *n* (%)	14 (14.58)	10 (14.49)	4 (14.81)	0.967
CKD (eGFR < 30 mL/min), *n* (%)	2 (2.08)	2 (2.9)	0	0.371
Previous DVT/PE, *n* (%)	6 (6.25)	4 (5.8)	2 (7.41)	0.769
Chronic Anemia, *n* (%)	4 (4.17)	2 (2.9)	2 (7.41)	0.320
COPD/Asthma, *n* (%)	10 (10.42)	9 (13.04)	1 (3.7)	0.178
Cancer, *n* (%)	10 (10.42)	10 (14.49)	0	0.036
Liver disease, *n* (%)	3 (3.13)	3 (4.35)	0	0.270
Severe Cognitive impairment, *n* (%)	7 (7.29)	5 (7.25)	2 (7.41)	0.978
Immunodepression, *n* (%)	5 (5.21)	4 (5.8)	1 (3.7)	0.678
Administered drugs				
Hydroxychloroquine, *n* (%)	63 (67.74)	48 (71.64)	15 (67.59)	0.193
Tocilizumab, *n* (%)	40 (41.67)	30 (43.48)	10 (37.04)	0.564
Azithromycin, *n* (%)	23 (23.96)	18 (26.09)	5 (18.52)	0.434
Darunavir/Cobicistat, *n* (%)	7 (7.29)	3 (4.35)	4 (14.81)	0.076
Anticoagulants (therapeutic dose), *n* (%)	21 (21.88)	13 (18.84)	8 (29.63)	0.250
Clinical features (at the moment of the ultrasound)				
Respiratory features				
qCSI, mean (SD)	2.12 (2.89)	0.5 (0.86)	6.25 (2.01)	<0.001
Respiratory rate, mean (SD), breaths/minute	20.29 (5.26)	18.81 (4.2)	24.07 (5.86)	<0.001
SpO2, mean (SD), %	94.07 (4.81)	95.42 (2.43)	90.62 (7.22)	0.002
P/F, mean (SD), ratio	290.89 (89.66)	330.71 (60.3)	199 (78.49)	<0.001
pO2, mean (SD), mmHg	75.23 (41.56)	69.65 (12.74)	88.11 (72.4)	0.208
Diffuse lung crepitations, *n* (%)	8 (8.33)	3 (4.41)	5 (18.52)	0.025
Bronchostenosis, *n* (%)	0 (0)	0	0	
Pathological chest X-ray (*n* = 88), *n* (%)	75 (85.23)	51 (82.26)	24 (92.31)	0.225
Pneumothorax, *n* (%)	0 (0)	0	0	
Pleural Effusion, *n* (%)	35 (36.6)	20 (28.99)	15 (55.56)	0.0150
*Other physical examination features*				
Body temperature, mean (SD), Celsius	36.57 (2.46)	36.33 (2.77)	37.18 (1.24)	0.041
SBP, mean (SD), mmHg	132.6 (18.08)	130.42 (16.5)	138.18 (20.9)	0.091
Heart rate, mean (SD), beats/minute	88.44 (17.42)	88.31 (17.11)	88.77 (18.52)	0.911
Dilated IVC (*n* = 33), *n* (%)	2 (6.06)	0	2 (20)	0.026
Reduced EF (*n* = 26), *n* (%)	4 (15.38)	3 (18.75)	1 (10)	0.547
*Laboratory tests*				
Lactate (*n* = 69), mean (SD), mmol/L	1.35 (0.88)	1.31 (0.91)	1.46 (0.82)	0.509
Hemoglobin, mean (SD), g/L	134.71 (18.88)	135.5 (18.9)	132.7 (19.04)	0.518
Neutrophils, mean (SD), cells/mm^3^	4857.84 (3130.76)	4278.75 (2474.57)	6267.82 (4063.33)	0.036
Platelets, mean (SD), cells/mm^3^	225,555.2 (88,245.85)	215,277.18 (81,303.13)	251,821.25 (100,840.02)	0.100
Limphocytes, mean (SD), cells/mm^3^	1517.97 (1172.77)	1473.14 (955.24)	1646.25 (1730.68)	0.706
Creatinine, mean (SD), µmol/L	78.74 (21.87)	80.41 (22.25)	74.47 (20.63)	0.220
D-dimer (*n* = 70), mean (SD), µg/L	1680.88 (4549.48)	732.04 (1148.74)	4053 (7966.81)	0.078
LDH, mean (SD), U/L	333.84 (128.73)	308.6 (113.57)	390.85 (144.29)	0.012
CRP, mean, (SD), mg/L	67.56 (68.51)	48.53 (44.93)	117.82 (92.24)	0.001
PCT (*n* = 62), mean, (SD), ng/mL	0.58 (3.18)	0.59 (3.61)	0.56 (1.87)	0.963
BNP (*n* = 51), mean, (SD), pg/mL	888.74 (4888.88)	1246.83 (6266.91)	333.8 (593.76)	0.426
Troponin I (*n* = 73), mean, (SD), ng/L	12.13 (14.73)	9.96 (11.52)	17.52 (20)	0.116
Ferritin (*n* = 64), mean, (SD), ng/mL	849.03 (788.7)	765.36 (744.74)	1062.83 (877.35)	0.214
CPK (*n* = 74), mean, (SD), U/L	151.04 (270.86)	139.87 (273.13)	183.36 (268.79)	0.549
ALT, mean, (SD), U/L	46.18 (41.28)	41.33 (28.49)	58.5 (62.23)	0.186
AST, mean, (SD), U/L	46.2 (28.38)	41.34 (20.61)	58.53 (40.08)	0.045
Clinical outcomes				
Oxygen rate flow ≥ 12 L/min, *n* (%)	28 (29.16)	14 (20.28)	14 (51.85)	0.002
NIV/OTI, *n* (%)	24 (25)	13 (18.84)	11 (40.74)	<0.001
Death, *n* (%)	8 (8.33)	4 (5.79)	4 (14.81)	0.150

BMI: Body Mass Index; TIA: Transitory Ischemic Attack; CKD: Chronic Kidney Disease; DVT/PE: Deep Venous Thrombosis/Pulmonary Embolism; COPD: Chronic Obstructive Pulmonary Disease; NIV: Non Invasive Ventilation; OTI: Oro Tracheal Intubation; qCSI: quick COVID Severity Index; SBP: Systolic Blood Pressure; SpO_2_: Pulse oxymetry; P/F: pO_2_/FiO_2_; pO_2_: arterial oxygen partial; FiO_2_: fractional inspired oxygen; LDH: Lactate-Dehydrogenase; CRP: C-Reactive Protein; PCT: Procalcitonin; BNP: Brain Natriuretic Peptide; CPK: Creatin Phosphokinase; ALT: Alanine Aminotransferase; AST: Aspartate Aminotransferase; IVC: Inferior Vena Cava; EF: Ejection Fraction of the left ventricle.

## Data Availability

The datasets during and/or analyzed during the current study are available from the corresponding author on reasonable request.

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
