# Peer review of "Lung Ultrasound Patterns and Clinical-Laboratory Correlates during COVID-19 Pneumonia: A Retrospective Study from North East Italy"

_jcm, 2021, doi:10.3390/jcm10061288_

Round 1

Reviewer 1 Report

The significance of result could be improved.

The authors state in the Abstact: „While consolidations correlated more with inflammatory biomarkers, B-lines correlated more with end-organ damage, including extrapulmonary involvement.“ This point should be more outlined in abstract and discussion.

The simple severity index has been developed and validated to identify patients who will progress to respiratory failure within 24 h, and it is based on a few rou[1]tinely available measures. It very important for imaging diagnosis and monitoring.

In disussicion section is the value of LUS in Thromboembolism (> 30% in COVID-19) not mentioned. It is an important reason for mortality.

Author Response

To

Reviewer 1

March 17, 2021

RE: Manuscript ID jcm-1159544

Dear,

Thank you for the opportunity to revise and resubmit our manuscript, entitled "Lung ultrasound patterns and clinical-laboratory correlates during COVID-19 pneumonia. A retrospective study from North East Italy". We have addressed your comments in the revised manuscript (using track changes) and in the point-by-point response below.

We hope that you will favourably consider the revised manuscript.

- - - - -

Response to Comments:

Point 1: The significance of result could be improved.

Response 1: We enlarged the discussion section, addressing with more details the clinical implications of our data.

Point 2: The authors state in the Abstract: „While consolidations correlated more with inflammatory biomarkers, B-lines correlated more with end-organ damage, including extrapulmonary involvement.“ This point should be more outlined in abstract and discussion.

Response 2: We added a brief statement in the Abstract, to specify the clinical implications of our data; moreover, we enlarged the discussion section, addressing with more details the clinical implications of our data.

Point 3: The simple severity index has been developed and validated to identify patients who will progress to respiratory failure within 24 h, and it is based on a few routinely available measures. It very important for imaging diagnosis and monitoring.

Response 3: We thank the reviewer to have underlined this concept. We feel that the possibility of stratifying the risk of a patient with easy tools is of paramount importance in clinical practice.

Point 4: In discussion section is the value of LUS in Thromboembolism (> 30% in COVID-19) not mentioned. It is an important reason for mortality.

Response 4: We agree that thromboembolism is a common complication of COVID-19 patients; however, we believe that lung ultrasound is of limited value for diagnosing this condition in our patients. We addressed the issue in the discussion section.

- - - - -

We really appreciate your effort and your kind help to improve this manuscript; the new version is attached.

Thank you again for the opportunity to revise and resubmit this article for consideration by Journal of Clinical Medicine. My co-authors and I look forward to hearing from you.

Sincerely,

Dr. Riccardo Senter on behalf of the authors

Department of Medicine,

University of Padova, 

Via N. Giustiniani 2, 35128, Padova, Italy

E-mail: riccardo.senter@aopd.veneto.it

Reviewer 2 Report

Dear Authors

Sincere congratulations and admiration. Your publication is incredibly interesting, innovative, and scientifically combines modern diagnostic imaging with "living" clinical practice.

The graphic form of presentation of the results you choose is very clear and attractive. At this point, I have one of two remarks - the ROC curve in Figure 1C is invisible in the version presented to me for review. Therefore, this element requires a graphic correction.

The conclusions you present are brief and very conservative. However, the real value of this work is, in my opinion, the correlation analysis of the COVID-19 ultrasound image and the biochemical and respiratory parameters.

My review is short, but I read the article very carefully.

In addition to the one presented above, I submit one more proofreading note - the reference entry requires correction (in the last section of the article, its number appears twice for each reference).

I select the option - Accept after minor only on the basis of the two points indicated above, in which the text requires editing.

Congratulations once again.

Author Response

To

Reviewer 2

March 17, 2021

RE: Manuscript ID jcm-1159544

Dear,

Thank you for the opportunity to revise and resubmit our manuscript, entitled "Lung ultrasound patterns and clinical-laboratory correlates during COVID-19 pneumonia. A retrospective study from North East Italy". We really thank you for the gratifying words and the high opinion on the value of this article.  We have addressed your comments in the revised manuscript (using track changes) and in the point-by-point response below.

We hope that you will favourably consider the revised manuscript.

- - - - -

Response to Comments:

Point 1: Sincere congratulations and admiration. Your publication is incredibly interesting, innovative, and scientifically combines modern diagnostic imaging with "living" clinical practice.

Response 1: We thank you again for your kindness and for the gratifying words. We are also happy to share with you the interest of looking for new evidences that could be immediately applied to the common clinical practice.

Point 2: The graphic form of presentation of the results you choose is very clear and attractive. At this point, I have one of two remarks - the ROC curve in Figure 1C is invisible in the version presented to me for review. Therefore, this element requires a graphic correction.

Response 2: We are extremely sorry for the editing problems in the figure of the previous version of the manuscript. In the revised manuscript the figure is corrected and we believe improved.

Point 3: The conclusions you present are brief and very conservative. However, the real value of this work is, in my opinion, the correlation analysis of the COVID-19 ultrasound image and the biochemical and respiratory parameters.

Response 3: We thank you again for your high opinion on the value of our data. We enlarged the discussion and the conclusion sections, addressing with more details the clinical implications and the importance of our data.

Point 4: My review is short, but I read the article very carefully. In addition to the one presented above, I submit one more proofreading note - the reference entry requires correction (in the last section of the article, its number appears twice for each reference).

Response 4: We really thank you for your careful assessment and for helping us to improve our work. We amended the misprints.

- - - - -

We really appreciate your effort and your kind help to improve this manuscript; the new version is attached.

Thank you again for the opportunity to revise and resubmit this article for consideration by  Journal of Clinical Medicine. My co-authors and I look forward to hearing from you.

Sincerely,

Dr. Riccardo Senter on behalf of the authors

Department of Medicine,

University of Padova, 

Via N. Giustiniani 2, 35128, Padova, Italy

E-mail: riccardo.senter@aopd.veneto.it
